# Enhanced Antibacterial and Anti-Inflammatory Activities of the Combination of *Cannabis sativa* and Propolis Extracts: An In Vitro Study

**DOI:** 10.3390/ijms262211181

**Published:** 2025-11-19

**Authors:** Naruemon Perstwong, Wongsakorn Phongsopitanun, Visarut Buranasudja, Sornkanok Vimolmangkang

**Affiliations:** 1Graduate Program in Pharmaceutical Science and Technology, Faculty of Pharmaceutical Sciences, Chulalongkorn University, Bangkok 10330, Thailand; 6471003333@student.chula.ac.th; 2Department of Biochemistry and Microbiology, Faculty of Pharmaceutical Sciences, Chulalongkorn University, Bangkok 10330, Thailand; wongsakorn.p@chula.ac.th; 3Department of Pharmacology and Physiology, Faculty of Pharmaceutical Sciences, Chulalongkorn University, Bangkok 10330, Thailand; visarut.b@pharm.chula.ac.th; 4Research Cluster for Cannabis and Its Natural Substances, Chulalongkorn University, Bangkok 10330, Thailand; 5Center of Excellence in Plant-Produced Pharmaceuticals, Faculty of Pharmaceutical Sciences, Chulalongkorn University, Bangkok 10330, Thailand; 6Department of Pharmacognosy and Pharmaceutical Botany, Faculty of Pharmaceutical Sciences, Chulalongkorn University, Bangkok 10330, Thailand

**Keywords:** CBD, hemp extract, cannabinoids, combination therapy, additive effect

## Abstract

Sore throat, commonly associated with pharyngitis and tonsillitis, is primarily caused by bacterial pathogens. Conventional therapies rely on antibiotics and anti-inflammatory drugs; however, concerns about adverse effects, antibiotic resistance, and drug interactions have encouraged the search for alternative remedies. *Cannabis sativa* L. (CS) has demonstrated potential in relieving sore throat and inflammation, while propolis, a bee-derived natural product, exhibits notable antibacterial and anti-inflammatory activities. This study aimed to investigate the enhanced antibacterial and anti-inflammatory effects of combining CS and propolis extracts (PE). Results found that CS and PE exhibited antibacterial effects against *Streptococcus pyogenes* DMST 4369, *Staphylococcus aureus* ATCC 25923, and *Pseudomonas aeruginosa* ATCC 9027. Their combination produced additive antibacterial effects against *S. pyogenes* and *S. aureus*. Cannabidiol (CBD) was identified as an active antibacterial constituent against *S. pyogenes*. Additionally, the PE-CBD in CS solution combination at concentration 625:0.125 µg/mL significantly reduced NO production and suppressed proinflammatory cytokines in LPS-stimulated macrophages. This study highlights the enhanced antibacterial and anti-inflammatory potential of the *C. sativa* and propolis combination, emphasizing the need to verify synergistic effects and determine the appropriate ratio for rational product development. Further research is needed to investigate the underlying mechanisms of action, particularly the anti-inflammatory pathways, in animal models. In addition, studies on hepatotoxicity should be conducted to ensure safety.

## 1. Introduction

Sore throat, a common symptom associated with pharyngitis and tonsillitis, is often caused by pathogens such as *Streptococcus pneumoniae* and *Streptococcus pyogenes*. Symptoms of a sore throat, including redness, swelling, and inflammation, represent the body’s immune response to these pathogens [1]. Conventional treatment for a sore throat involves antibiotics and anti-inflammatory agents for pain relief. However, the limitations of modern medicine, including adverse effects, increased antibiotic resistance, and potential drug interactions, have led to an increased interest in alternative therapeutic options [2]. The Thai Traditional and Alternative Medicine Department (DTAM), Ministry of Public Health, Thailand, recognizes the potential of herbal medicines as alternatives to non-steroidal anti-inflammatory drugs (NSAIDs) [3].

*Cannabis sativa* (CS) or hemp has been integrated into Thai Traditional medicine (TTM) in particularly in formulations such as Prab-Chom-Poo-Tha-Weeb, which are used to treat sore throat and common colds [4]. Historically, TTM doctors employed CS leaves to enhance appetite and treat asthma, although excessive use could cause adverse effects, such as fear, dizziness, and hallucinations. CS flowers were traditionally used to manage psychosis, promote sleep, and stimulate appetite [5]. CS notably contains various phytocannabinoids [6]. In previous studies, CBD and *C. sativa* extract demonstrated analgesic effects through cannabinoid receptors and serotoninergic 5-HT1a and exhibited chronic pain relief, palliative care, analgesic, and anti-SARS-CoV-2 properties [7,8,9,10]. Furthermore, CBD, cannabinol (CBN), cannabigerol (CBG), and delta-9-tetrahydrocannabinol (Δ9-THC) have been shown to have antioxidant properties [11]. In addition, terpenoids (caryophyllene) exhibited antidepressant, antinociceptive, and anti-inflammatory properties [6]. Methanolic hemp leaf extract also revealed antirheumatoid activity by inhibiting nitric oxide (NO) and prostaglandin E2 (PGE2) production [12]. Recent studies have shown that cannabinoids, such as CBD, CBCA, CBG, CBGA, THCV, Δ8-THC, and exo-THC, exhibit antibacterial activity, particularly against *Staphylococcus aureus* and *S. pyogenes* [13]. In addition, CBD has been demonstrated to have an additive anti-inflammatory effect when used in combination with moringin in LPS-activated mouse macrophages. These findings suggest that the combination of low-dose CBD with moringin could offer a promising therapeutic approach for the treatment of inflammatory diseases [14].

Propolis, a natural resinous substance produced by bees, has traditionally been employed in ethnomedicine to treat gastrointestinal diseases, as a mouth disinfectant, and for wound healing [15]. Propolis is a complex natural substance comprising various bioactive compounds, including prenylated cinnamic acids, prenylflavonoids, and flavonoids. In Thailand, propolis derived from *Styrax* trees has been found to contain novel phenylallylflavanones, specifically (7″S)-8-[1-(4′-hydroxy-3′-methoxyphenyl)prop-2-en-1-yl]-(2S)-pinocembrin and (E)-cinnamyl-(E)-cinnamylidenate [16]. Currently, propolis extract exhibits significant anti-inflammatory effects by inhibiting the mRNA expression of IL-1β, IL-6, and cyclooxygenase-2 (COX-2) in mouse macrophage cells and demonstrates anti-breast cancer activity [17,18]. In addition, flavonoids (solophenols B, solophenols C, and solophenols D) isolated from propolis originating from the Solomon Islands have shown antibacterial effects against *S. aureus* and *Pseudomonas aeruginosa* [19]. Moreover, propolis has been reported to be used in combination. For instance, propolis demonstrated a synergistic antibacterial effect against *Escherichia coli*, *S. aureus* and *Candida albicans* when used with honey or ethyl alcohol, and the antimicrobial effect of propolis depends on its various geographical origins. Its antibacterial activity may be attributed to chemical compounds such as phenolics and flavonoids, including pinocembrin and galangin [20]. A study investigated the biological effects of propolis derived from honeybees that had access to cannabis plants. The findings suggest that cannabis may enhance the antimicrobial properties of propolis [21].

Owing to their well-established pharmacological properties—such as pain relief, anti-inflammatory, antibacterial, and immune-modulating effects—both *propolis* and *C. sativa* (medicinal hemp) extracts have demonstrated promising therapeutic potential. Propolis has long been used for sore throat relief, particularly during the COVID-19 pandemic, due to its potent antimicrobial activity. Meanwhile, *C. sativa* is increasingly recognized for its medicinal applications, including inflammation control and immune regulation. Interestingly, a previous study reported that propolis collected from bees foraging on cannabis plants contained cannabinoids such as cannabidivarinic acid, Δ^9^-tetrahydrocannabinolic acid, and cannabidiolic acid, and exhibited stronger antimicrobial activity than propolis from non-cannabis plants [21]. However, these findings remain limited by variability in bee foraging behavior, cannabis cultivars, and bee species.

Conventional throat spray formulations containing propolis are primarily used to relieve throat irritation and exert mild antibacterial effects. By combining propolis with *C. sativa* extract, this study aimed to broaden the indication of throat spray formulations beyond antimicrobial action to include anti-inflammatory benefits, thereby providing more comprehensive relief for sore throat and upper respiratory tract infections.

To date, no study has directly investigated the deliberate combination of *C. sativa* and propolis extracts, nor optimized their concentration ratio, despite assumptions that such pairing could potentiate their biological activities. Therefore, this study developed and evaluated a novel formulation combining *C. sativa* extract—the major active ingredient in sublingual oil products for insomnia and chronic pain relief—with propolis extract, the major active ingredient in commercially available throat sprays. The study explored potential additive or synergistic antibacterial and anti-inflammatory interactions, identified major active compounds contributing to antibacterial activity, and investigated anti-inflammatory mechanisms through cytokine analysis.

By experimentally verifying these biological effects and assessing concentration-dependent safety profiles, this research provides the first scientific evidence supporting a standardized cannabis–propolis formulation with enhanced therapeutic efficacy. This innovative approach establishes a promising foundation for the development of safe, effective cannabis-based medicinal products and expands the application of natural therapeutic combinations in modern pharmaceutical development.

## 2. Results

### 2.1. HPTLC-Based Preliminary Chemical Profiling of C. sativa and Propolis Extract

Quality control of herbal medicines, particularly for crude drugs (raw plant material) and final preparations, remains a considerable challenge. HPTLC offers a simple and reliable method for authenticating the plant materials [22]. In this study, HPTLC was employed following previous reports to identify the chemical profiles of *C. sativa* extract (CS^ext^) [11,23] and PE [24]. The results are shown in Figure 1. Cannabinoids, including Δ9-THC and CBD, were detected at R_f_ 0.33–0.40 under observation with white light and the fast blue B salt TS. These observations corresponded with the recommended TLC/HPTLC method from United Nations Office on Drugs and Crime (UNODC) and HPTLC Association, although the R_f_ values showed slight variation [25,26]. The chemical profile of PE represented by gallic acid (GA) at R_f_ 0.33 was observed under UV at 254 nm. This is a preliminary quality assessment of raw materials containing CS^ext^ and PE before they are used in product formulation. This developed system was applied to investigate the bioactive compounds in the crude extract for antibacterial activity using the TLC-bioautography technique.

### 2.2. CS and PE Combination Exhibits an Additive Effect on Antibacterial Activity

The antibacterial activity of propolis extract solution (PE^sol^) and *C. sativa* extract solution (CS^sol^) was evaluated using the agar disk diffusion method, and the results are presented in Table 1. The experimental results demonstrate that PE^sol^ at 20% and 100% *v*/*v* exhibited inhibitory effects on the growth of *S. pyogenes*, *S. aureus*, and *P. aeruginosa*, with inhibition zone values ranging from 12.00 to 15.00 mm. Notably, the 20% *v*/*v* PE^sol^ revealed the largest inhibition zone against *P. aeruginosa*. A comparison between the 20% and 100% *v*/*v* PE^sol^ concentrations revealed no significant difference in the size of the inhibition zones. Therefore, the 20% *v*/*v* PE^sol^ was selected for further MIC/MBC investigations. Meanwhile, 20 mg/mL CS^ext^, containing 11.44 mg/mL of CBD, showed inhibition zones of 13.00 to 17.00 mm against *S. pyogenes and S. aureus*, with the largest inhibition zone observed against *S. pyogenes*. The diluents 10% DMSO (*v*/*v*) and sterile water were used as negative controls. Our results indicated that CS^ext^ and PE^sol^ demonstrated antibacterial activity. However, their efficacy was lower than that of the positive controls, except for PE^sol^, which exhibited a zone of inhibition against *P. aeruginosa* larger than that of amoxicillin and similar to that of chloramphenicol.

We further assessed their efficacy using the MIC/MBC method. The results revealed that PE^sol^ and CS^sol^ exhibited antibacterial activity against Gram-positive and Gram-negative bacterial strains (Table 2). PE^sol^ showed similar MIC/MBC values (1.25–5.00% *v*/*v*) against Gram-positive and Gram-negative bacteria. However, CS^sol^ showed lower values against Gram-positive bacteria than against Gram-negative bacteria.

### 2.3. The Optimization of Combination Between CS and PE

The combination of CS and PE was initially prepared at a simple 1:1 ratio, reflecting the traditional practice commonly found in TTM formulations [5], where equal proportions are often used for ease of preparation and practical application. However, due to regulatory restrictions in Thailand, where CS is classified as a controlled substance permitted only for medical use, the ratio of CS in the mixture was subsequently reduced to 0.5. Furthermore, this study aimed to designate PE as the major constituent in the formulation. As a result, only the concentration of CS was varied based on the MIC results. This adjustment was made to ensure compliance with national legislation, which mandates that cannabis-derived products be used at minimal and strictly regulated dosages to ensure safe therapeutic use.

Based on the MIC results of the individual extracts, PE^sol^ and CS^sol^ were combined in ratios of 1:1 MIC and 1:0.5 MIC to assess their potential additive effects. The antibacterial activity of the mixtures was evaluated according to the guidelines established by EUCAST [27]. Table 3 shows the results of the checkerboard dilution of the mixtures. Combination 1 (ratio 1:1 MIC) exhibited an additive effect (FIC index = 1) against *S. pyogenes*, whereas Combination 2 (ratio 1:0.5 MIC) demonstrated an additive effect against *S. aureus*. An additive effect of a combination, according to EUCAST, is interpreted as one in which the effect of the combination is equal to the sum of the effects of the individual components. An additive antibacterial effect was observed upon adjustment of CS^sol^ concentration, with efficacy varying by bacterial strain. In contrast, both combinations did not exhibit an effect against Gram-negative bacteria, specifically *P. aeruginosa* and *K. pneumoniae*. Interestingly, this finding revealed that the combination only produced an additive antibacterial effect against Gram-positive bacteria.

The interpretation of the FIC index was as follows: an FIC index ≤ 0.5 indicates a synergistic effect, while an FIC index between 0.5 and 1 indicates an additive effect. An FIC index ranging from 1 to 2 indicates indifference, while an FIC index > 2 indicates antagonism; EUCAST [27].

The combination exhibited stronger additive antibacterial effects against Gram-positive bacteria compared to Gram-negative strains. Notably, the MIC value of the combination against *S. pyogenes* was lower than that against *S. aureus*. Therefore, *S. pyogenes* was selected as the target strain for the TLC-bioautography assay due to its higher susceptibility and clear response to the test compounds. To confirm the antibacterial activity of the extracts, we employed TLC-direct bioautography to screen and identify the active compounds. Figure 2 illustrates the TLC-direct bioautography, which revealed that CBD and Δ9-THC present in CS^ext^ demonstrated antibacterial effects against *S. pyogenes*, with inhibition zones corresponding to R_f_ values of 0.33 and 0.40, respectively. However, no detection was observed for 100 mg/mL PE.

### 2.4. Cytotoxicity

The cytotoxic activity of PE, CS^sol^, and CBD was evaluated using the MTT assay. In addition, the cytotoxicity of vehicles, PG and LPS, was also assessed, the corresponding data are presented in Appendix A. The results showed that concentrations up to 2.5 mg/mL of propolis extract (PE) (Figure 3B), CBD equivalent to 1 µg/mL in *C. sativa* solution (CS^sol^) (Figure 3A), and 1 µM (~314 µg/mL) of CBD (Figure 3C) maintained RAW264.7 cell viability above 80%. Notably, the concentration of *C. sativa* solution required to achieve an effect equivalent to CBD was lower than that of CBD alone. This observation suggests that other cannabinoids present in the *C. sativa* solution may contribute to its biological activity and potentially enhance its safety profile in RAW264.7 cells. Therefore, those concentrations were selected for the combinations. Two combinations were evaluated: the first consisting of 2.5 mg/mL PE and CBD eq. to 1 µg/mL in CS^sol^, and the second consisting of 2.5 mg/mL PE and 1 µM (~314.47 µg/mL) CBD, mixed at two ratios of 1:1 and 1:0.5. Combination 1 (PE and CS^sol^) at ratios of 1:1 and 1:0.5 was non-toxic toward RAW264.7 cells at concentrations up to 625.00 µg/mL PE and corresponding concentrations of CBD in CS^sol^ (0.250 µg/mL and 0.125 µg/mL, respectively) (Figure 4A,B). Similarly, combination 2 of PE and CBD at both ratios showed no cytotoxicity at the same concentrations of PE and CBD (Figure 4C,D). These concentrations were identified as the highest non-toxic doses, indicating their suitability for further evaluation of anti-inflammatory activity. Therefore, PE at a concentration of 625.00 µg/mL, and CBD at concentrations equivalent to 0.250 µg/mL and 0.125 µg/mL in CS^sol^ and CBD, respectively, which were identified as non-cytotoxic in the combination treatments, were selected for further evaluation of their anti-inflammatory potential. This was assessed by measuring the inhibition of NO production and proinflammatory cytokines, as presented in Figure 4E,F and Figure 5.

### 2.5. CS and PE Combination Significantly Reduced NO Production and Suppressed the Release of Proinflammatory Cytokines

The anti-inflammatory effects of the two combinations—PE with CS^sol^ or CBD—were evaluated by quantifying the NO levels in the media of the RAW264.7 cells. The NO concentrations were calculated using the equation obtained from the standard curve of nitrite, as shown in Appendix A. Our findings indicated that both combinations exhibited a dose-dependent decrease in NO production compared with the LPS-treated group as represented in Figure 4E,F. Overall, the combinations demonstrated a significantly lower release of NO than the individual treatments. As shown in Figure 4E, a high concentration as 626 µg/mL PE mixed with CS^sol^ eq. to CBD 0.125 µg/mL significantly reduced NO production when compared to LPS group. Notably, this concentration—identified as the maximum non-toxic dose—also resulted in a significantly greater reduction in NO production than either PE or CS^sol^ alone at the same concentration. Furthermore, the combination of PE with CBD was more effective in suppressing LPS-induced NO production than the combination of PE with CS^sol^, suggesting a stronger anti-inflammatory potential. These results suggest the potential anti-inflammatory effect of PE relative to both CS^sol^ and CBD. The production of cytokines IL-6, IL-8, and TNF-α was determined using ELISA kits. Their levels were calculated based on the equations obtained from the standard curves, as shown in Appendix A. Interestingly, the combination of PE:CS^sol^ eq. to CBD (625:0.125 µg/mL) significantly suppressed the levels of IL-6, IL-8, and TNF-α, compared to the control group (# *p* < 0.05), as indicated by the ELISA results, which correlate with the observed NO production (Figure 5). Especially, the level of IL-8 observed in the combination group is significantly lower than the 625 μg/mL PE-only group († *p* < 0.05).

## 3. Discussion

For the identification of the chemical profile of *C. sativa*, it was noted in the THP that CBD and THC were selected for standardized hemp female flower using the TLC method [28]. While propolis had no official standardization for identification in the pharmacopeia, its TLC chemical profile and derivatizing mobile phase were reported. Chrysin, kaempferol, p-coumaric acid, apigenin, caffeic acid, quercetin, and GA were used as reference standards for the identification of the Serbian PE [29]. GA has demonstrated antibacterial activity against *S. aureus* and *P. aeruginosa* [30]. In addition, it is commonly used as a reference standard in THP for the identification of *Terminalia chebula* gall and *Phyllanthus emblica* [31,32]. Therefore, to identify CS^ext^ and PE using the HPTLC technique, CBD and THC, and GA were used as chemical markers for CS^ext^ and PE, respectively. The chemical profiles of THC and CBD were present at R_f_ 0.33–0.40 with red-pinkish and orange-brownish zones, respectively, which were related to the TLC/HPTLC recommended method from UNODC and HPTLC Association, but the R_f_ was little shifted [25,26]. PE was present with GA at R_f_ 0.33. GA is a well-known phenolic compound with potent antioxidants and anti-inflammatory properties. It has been widely used as a standard marker for antioxidant activity, particularly in 2,2-Diphenyl-1-picrylhydrazylradical scavenging activity (DPPH) assays [33]. In this study, the presence of GA in PE suggests that PE may also exhibit anti-inflammatory activity. This investigation may be used to identify the raw material contained in CS^ext^ and PE before use in product preparation.

In the screening of the antibacterial activity of the individual extracts, we investigated whether both CS and PE^sol^ exhibit antibacterial activity. In pharmacology, comprehending the potential additive effects is crucial for assessing the efficacy of antibiotic combinations. A combination of medicinal compounds demonstrates an additive effect when the overall efficacy equals the sum of the individual effects of each component. This principle is important for optimizing therapeutic strategies and improving treatment outcomes [34]. Our results align with previous reports, suggesting that combining the two extracts could provide broader antibacterial coverage, effectively targeting both Gram-positive and Gram-negative bacteria. Our findings revealed that the MIC of PE^sol^ was 2.5% *v*/*v* or equivalent to 25 mg/mL for *S. aureus* and *S. pyogenes*, which indicates that the antibacterial activity of PE^sol^ falls within a similar range to those previously reported. Ethanolic PE demonstrated effectiveness against *S. aureus*, exhibiting an inhibition zone of 18.22 ± 1 mm and an MIC of 10 mg/mL [35]. A previous study indicated that the antibacterial activity of PE against *S. aureus*, *P. aeruginosa*, and *S. pyogenes* had MIC values of 6.25% *v*/*v* [34]. The findings indicate that the MIC of CS^sol^ was 1 µg/mL for *S. pyogenes* and 1.56 µg/mL for *S. aureus*, suggesting that the antibacterial activity of CS^sol^ falls within a similar range as that previously reported. *C. sativa* extracts, supercritical CO_2_ extracts, and water extracts of inflorescences exhibited antibacterial activities against *P. aeruginosa* and *S. aureus* [36,37]. CBD is the most effective cannabinoid that exhibits antibacterial activity against *S. aureus* and *S. pyogenes*. The MICs of CBD were 0.65–32 µg/mL for *S. aureus*, 0.5–4 µg/mL for MRSA, and 1–2 µg/mL for vancomycin-resistant *S. aureus* strains. Furthermore, CBG and Δ9-THC have demonstrated antibacterial efficacy against *S. pyogenes*, with MICs 0.6 and 5 mg/L, respectively [13]. However, the combination of CS^sol^ and PE^sol^ in Combinations 1 and 2, which contained low concentrations of CS^sol^ (0.5–0.78 µg/mL) and PE^sol^ (1.25% *v*/*v*), demonstrated an additive antibacterial effect against *S. pyogenes* and *S. aureus*. These results align with previous studies, which suggest that propolis, particularly when derived from honeybees that have access to cannabis plants, may enhance its beneficial properties. Notably, our findings demonstrated more promising outcomes compared to previous studies, showing high antibacterial efficacy of the propolis extracts obtained from cannabis. However, no significant difference was observed in the MIC against *S. aureus* compared with the control group [21], the propolis derivative from the cannabis plant still has certain limitations, such as variations in raw material sources and the complexity of controlling and developing formulation ratios in the future. Nevertheless, our study indicates that an optimal combination ratio enhances antimicrobial activity through an additive effect. Furthermore, determining the appropriate ratio of the combination of propolis extract and *C. sativa* extract could facilitate future investigations into their additive antimicrobial effects, which appear to be more pronounced against Gram-positive than Gram-negative bacterial strains. In addition, the combination of PE and *C. sativa* extract enhanced antibacterial activity compared to each extract used individually. The overall data suggests that the interaction between CS^sol^ and PE^sol^ could contribute to a more potent antimicrobial effect, which could have important implications for the development of more effective natural antimicrobial agents.

Based on the TLC-bioautography results, our findings revealed that the antibacterial activity of CS^ext^ is attributed to the presence of CBD and Δ9-THC. CBD inhibited various bacteria, including *S. pyogenes*, *S. aureus*, MRSA, and *P. Aeruginosa* [38,39]. Given its consistent efficacy against various bacterial pathogens, CBD appears to be a key bioactive compound that is responsible for the antimicrobial properties of the *C. sativa* crude extract. This supports the potential of CBD as a natural antimicrobial agent, which may be valuable for future pharmaceutical or therapeutic applications. In addition, CBG has demonstrated efficacy against *S. pyogenes*, with MIC values ranging from 0.6 to 50 mg/L [13]. It should be noted that CBG could be considered as another reference standard for TLC/HPTLC analyses. The TLC-direct bioautography analysis of PE and GA did not reveal any antibacterial activity against *S. pyogenes*. Although gallic acid has been reported to exhibit antibacterial properties, no inhibitory effect against *S. pyogenes* was observed in this study at 1 mg/mL, 4 µL loaded on the TLC plate. Similarly, the crude propolis extract, tested at 100 mg/mL (4 µL), failed to produce any detectable inhibition zone in the bioautography assay. These findings suggest that *S. pyogenes* may be less susceptible to gallic acid and propolis at low concentration. In addition, the absence of inhibition could also be attributed to the compound instability during TLC development. As relevant to this report, gallic acid at concentrations below 1.88% *w*/*v* failed to produce an inhibition zone, supporting previous findings that *S. pyogenes*, a Gram-positive bacterium, exhibits resistance to gallic acid [30]. However, the lack of antibacterial activity observed in this study may be attributed to limitations in the separation system, which utilized only gallic acid as the reference compound. Therefore, these findings suggest the need to explore alternative mobile phases to effectively separate other bioactive constituents in propolis extract—such as luteolin, quercetin, caffeic acid, pinocembrin, and pinobanksin—which have been reported to possess antibacterial properties [29,40]. Ultimately, gallic acid also holds potential as a fundamental chemical marker due to its well-documented antioxidant and anti-inflammatory activities. Moreover, gallic acid is commonly used as a chemical marker in other plant species within the Thai Herbal Pharmacopeia (THP). Meanwhile, CBD and THC have been successfully utilized as biomarkers for detecting hemp flower extracts.

To determine the anti-inflammatory effect of the combined treatments, the production of NO was measured in two combinations: PE and CS^sol^ or CBD. Our findings demonstrated that the concentration of the two combinations of CBD and PE exhibited a dose-dependent decrease in the amount of NO produced in comparison to the LPS-treatment groups. Nevertheless, compared with treatment with CS^sol^ and PE alone, the combination with a low concentration of PE (625 μg/mL) and CS^sol^ equivalent to CBD (0.125 μg/mL) at a ratio of 1:0.5 demonstrated a significantly reduced release of NO. The combination of PE and CBD decreased the LPS-stimulated NO production more effectively than the combination of PE and CS^sol^ at similar concentrations. In addition, after LPS treatment, the levels of TNF-α and IL-1 decreased in response to treatment with CS extract and CBD [41]. CBD has been shown to downregulate the expression of proinflammatory markers, including TLR-4 and NF-κB transcription factors [14]. The anti-inflammatory effect of PE was also reported. Terpenoid-rich PE has been reported to inhibit NO production by downregulating key inflammatory mediators, including inducible nitric oxide synthase (iNOS), IL-1β, and IL-10 [42]. The ethanolic PE, enriched with flavonoids, has exhibited notable anti-inflammatory effects by inhibiting NF-κB activation and reducing TNF-α secretion [43]. Previous studies have highlighted the enhanced anti-inflammatory effects of combining CBD with moringin, which demonstrated superior results compared with the individual components. This combination notably improved inflammatory markers such as TNF-α and IL-10, as well as oxidative markers such as iNOS, nuclear factor erythroid 2-related factor 2 (Nrf2), and nitrotyrosine [14]. Similarly, our study revealed that the combination of PE-CS^sol^ at a concentration of 625:0.125 µg/mL significantly suppressed the levels of key inflammatory cytokines, including IL-6, IL-8, and TNF-α, compared with each extract used individually. Moreover, the combination of PE-CS^sol^ at the same concentration significantly reduced IL-8 and TNF-α levels compared with 625 µg/mL propolis extract (Figure 5B,C). However, no synergistic interaction was observed, indicating that the combination produced additive or enhanced effects rather than synergism. These consistent findings suggest that both combinations may exert their anti-inflammatory effects, at least in part, through a shared mechanism involving the downregulation of IL-6 and IL-8. Given the complex phytochemical profiles of both PE and CS^sol^, it is plausible that their synergistic effects involve multi-target pathways, similar to those observed in the CBD–moringin study. In line with previous studies, the anti-inflammatory effects of propolis appear to be mediated through multiple mechanisms. Evidence indicates that propolis downregulates key inflammatory signaling pathways, as TLR4 and NF-κB, leading to the suppression of proinflammatory cytokines such as IL-1β, IL-6, and TNF-α [44]. The combination of propolis extract with CBD or *C. sativa* extract demonstrated enhanced anti-inflammatory effects, particularly through a more significant reduction in nitric oxide production compared to individual treatments. These mechanisms correspond with our findings, in which PE alone significantly decreased IL-6 levels in LPS-stimulated macrophages, supporting its role as a multifunctional anti-inflammatory agent. However, a key limitation of this study is the inability to clearly characterize the nature of the interaction between propolis extract and *C. sativa* extract—whether it is synergistic or antagonistic. To address this gap, future research should employ the Chou–Talalay method using a broader range of combination concentrations, which would allow for a more precise and reliable assessment of their interaction dynamics. These findings emphasize the potential of this combination as a more effective anti-inflammatory agent, particularly in modulating critical pathways such as NF-κB and MAPK. This study contributes to the growing body of scientific evidence supporting the therapeutic potential of natural compounds, particularly propolis and cannabis extracts. The combination of these two extracts demonstrated an additive interaction in antibacterial activity against sore throat pathogens compared with the individual extracts. In addition, the combination exhibited enhanced interaction in anti-inflammatory responses when compared with the individual extracts. These findings highlight the potential of propolis and cannabis extracts as promising natural ingredients for further development in herbal pharmaceutical applications. To expand the therapeutic applicability of the combination, future studies should investigate its cytotoxicity in other cell lines, such as MRC-5 (human fibroblasts) or zebrafish models, which would provide broader insight into its safety profile.

## 4. Materials and Methods

### 4.1. Materials and Sample Preparation

A crude extract of *Cannabis sativa* (CS^ext^), derived from the inflorescence of the hemp strain Siam CA (Lot No. DeCsA110665.2) and containing 57.20% *w*/*w* CBD and 1.68% *w*/*w* THC, was sourced and analyzed by Leapdelab Co., Ltd., Samut Prakan, Thailand. The THC and CBD contents in CS^ext^ was quantitatively analyzed using high-performance liquid chromatography (HPLC) Appendix A. The propolis extract solution (PE^sol^) was obtained from Nova Health Co., Ltd. (Bangkok, Thailand), and the propolis was sourced locally in Thailand. All experimental procedures involving *C. sativa* were conducted under cannabis extraction and research licenses approved by the Thai Food and Drug Administration and the Thai Traditional and Alternative Medicine Department (DTAM), Ministry of Public Health, Thailand.

### 4.2. CS Sample Preparation

CS^ext^ was dissolved in DMSO (Cat. No. 445103) to obtain an initial solution of CS^ext^ containing 50 mg/mL. Hence, it was diluted to 1:10 in 60% *v*/*v* propylene glycol (PG) (Cat. No. 95904) and water to obtain the *C. sativa* solution (CS^sol^) containing 5 mg/mL CBD as the stock solution.

### 4.3. Propolis Sample Preparation

PE^sol^ is an ethanolic, water-based solution from Nova Health Co., Ltd. (Bangkok, Thailand). It was used directly for the antibacterial activity assay. For the anti-inflammatory activity assay, PE^sol^ was evaporated, freeze-dried, then collected the 14.95% *w*/*v* of PE. The PE was and resuspended in DMSO to prepare a 500 mg/mL PE stock solution.

### 4.4. High-Performance Thin-Layer Chromatography Analysis

The objective of the high-performance thin-layer chromatography (HPTLC) analysis was to investigate the chemical profile pattern of CS^ext^ and PE. According to the Thai Herbal Pharmacopeia (THP), CBD and THC (Cat. No. 13956-29-1 and Cat. No. THC-1098S-100, respectively; THC Pharm GmbH—The health concept, Frankfurt, Germany) are used as reference chemical markers for the qualitative analysis of cannabis female flowers [28]. PE is not listed in the THP. However, many chemical compounds used for the qualitative analysis of PE by HPTLC have been reported, including kaempferol, p-coumaric acid, caffeic acid, quercetin, and gallic acid (GA) (Cat. No. 149-91-7) [29]. According to a previous report, propolis was found to contain total phenolic compounds, including gallic acid. Gallic acid was also used as a reference standard for propolis extract, which exhibited antioxidative activity [33]. Therefore, this study selected CBD, THC, and GA as the reference standard markers for qualitative analysis. Ten milligrams of CS^ext^ and PE were dissolved in 1 mL of methanol, and 4 µL of the sample and 2 µL of the reference standards were applied onto the HPTLC plates using a semiautomatic applicator (CAMAG, Muttenz, Switzerland). The plates were developed in a pre-saturated chamber with the mobile phase vapor [A: toluene (Cas No. 108883)/diethylamine (Cas No. 8.03010.0500) (19.6:0.4); B: toluene/ethyl acetate (Cas No. 1.09623.2500): formic acid (Cas No. 64186) (6:5:1, *v*/*v*/*v*)] [45]. Development was performed manually, with a migration distance of 70 mm. Following heating at 100 °C for 2 min, the plates were sprayed with fast blue B salt TS (Cas No. 14263-94-6) and PEG400 (Cas No. 95904) for cannabinoid and gallic compound detection, respectively, and visualized under white and UV lights.

### 4.5. Antibacterial Activity

#### 4.5.1. Microorganisms, Culture Media, and Maintenance Protocols

The bacterial strains used in this study included *S. pyogenes* DMST 4369 (Department of Medical Sciences, Ministry of Public Health, Nonthaburi, Thailand), *S. aureus* ATCC 25923, *K. pneumoniae* ATCC 13883, and *P. aeruginosa* ATCC 9027 (ATCC, Manassas, VA, USA). The strains were cultured on Mueller–Hinton agar (MHA) (Cat. No. M391) at 37 °C for 24 h, with *S. pyogenes* incubated under CO_2_-enriched conditions. Before performing the antimicrobial assay, all bacterial cultures were adjusted to the 0.5 McFarland turbidity standard (Cat. No. R092), corresponding to approximately 1.5 × 10^8^ CFU/mL, using 0.9% (*w*/*v*) normal saline.

#### 4.5.2. Agar Disk Diffusion Method

The agar disk diffusion assay was performed in accordance with the guidelines of the Clinical and Laboratory Standards Institute (CLSI) [46]. Stock solutions of each test sample were prepared as follows: PE^sol^ was used at full strength (100% *v*/*v*) and also diluted to 20% (*v*/*v*) with sterile distilled water. For CS^ext^, the initial stock solution at 50 mg/mL was diluted with DMSO to a final concentration of 20 mg/mL CS^ext^, yielding a concentration equivalent to 11.44 mg/mL of CBD. A volume of 20 µL of each prepared sample was applied to 6 mm sterile paper disks (Cat. No. 2017-006, Whatman™, Marlborough, MA, USA). Negative controls included 10% (*v*/*v*) DMSO (used to dissolve CS^ext^) and sterile water (used to dilute PE^sol^). Chloramphenicol (Cas No. CT0013B, Thermo Scientific™ Oxoid™, Basingstoke, Hampshire, UK) and amoxicillin (Cas No. SD001, HiMedia^®^, Mumbai, India) were used as positive controls. The disks were placed onto Mueller–Hinton agar (MHA) plates previously inoculated with standardized bacterial suspensions, and the plates were incubated at 37 °C for 18–24 h. Following incubation, the diameters of the inhibition zones were measured to assess antimicrobial activity.

#### 4.5.3. Minimum Inhibitory Concentration and Minimum Bactericidal Concentration

The broth microdilution method was employed to determine the minimum inhibitory concentration (MIC) of the test samples, following the guidelines described in guidelines of the European Committee on Antimicrobial Susceptibility Testing (EUCAST) [47]. PE^sol^ was used at full strength (100% *v*/*v*) as the stock solution. Due to solubility limitations of CS^ext^ (20 mg/mL) in Mueller–Hinton broth (MHB) (Cat. No. M391, HiMedia^®^), it was substituted with CS^sol^, a soluble formulation containing 5 mg/mL of CBD, which served as the stock solution for this assay.

In a 96-well microtiter plate (Cat. No. 3599, Corning^®^, New York, NY, USA), 50 µL of a standardized bacterial suspension was added to each well, followed by 50 µL of the test sample, which had been prepared using two-fold serial dilutions in MHB. Sterility control wells contained MHB alone, while growth control wells contained MHB with the bacterial suspension in the absence of test compounds. To ensure that DMSO did not interfere with the antimicrobial assay, MHB containing 10% (*v*/*v*) DMSO was inoculated with each tested microorganism. The plates were incubated at 37 °C for 18–24 h. After incubation, the MIC was defined as the lowest concentration of the test sample that resulted in a visibly clear well, indicating complete inhibition of bacterial growth. To determine the minimum bactericidal concentration (MBC), 10 µL aliquots from wells showing no visible growth were plated onto Mueller–Hinton agar (MHA). Following overnight incubation, the MBC was recorded as having the lowest concentration at which no bacterial colonies were observed on the agar surface.

#### 4.5.4. Fractional Inhibitory Concentration and Index Using the Broth Microdilution Checkerboard Method

The combination of PE^sol^ and CS^sol^ was prepared using selected concentrations based on their individual MBC values specific to the bacterial strain. The stock solution of PE^sol^ was 100% *v*/*v* (4 × MBC), while the stock solution of CS^sol^ was 2 mg/mL (4 × MBC) as shown in Appendix A. The combinations were prepared at ratios of 1:1 and 1:0.5 according to the principles of Traditional Thai Medicine (TTM). The combinations were prepared as 2-fold serail dilution. The MIC values of the combinations were determined as described above. The synergy of the combinations was evaluated by calculating the fractional inhibitory concentration (FIC) index [27].

#### 4.5.5. Thin-Layer Chromatography-Bioautography

For thin-layer chromatography (TLC) analysis, 4 µL of 10 mg/mL CS^ext^ and 100 mg/mL PE were applied to aluminum TLC silica gel 60 F254 plates (CAMAG, Muttenz, Switzerland), following the conditions specified by the HPTLC method, and 2 µL of 5 µg/mL ampicillin (Cat. No. 69-52-3, Sigma Aldrich^®^, Darmstadt, Germany) was used as the positive control. The TLC plate was immersed in a bacterial suspension prepared in the MHB medium for 5 s and subsequently placed onto the MHA medium. The plate was incubated for 18 h, sprayed with 0.2% (*v*/*v*) INT dye (Cat. No. 58030, Sigma Aldrich^®^, Darmstadt, Germany), and incubated for 30 min. Inhibition zones were observed as clear bands around a red-purple background.

#### 4.5.6. Anti-Inflammatory Activity

Cytotoxicity

RAW 264.7 mouse macrophages were purchased from the American Type Culture Collection (ATCC, Manassas, VA, USA). Cells were cultured in Dulbecco’s Modified Eagle Medium (DMEM) (Cat. No. 12491015), supplemented with 10% *v*/*v* fetal bovine serum (Cat. No. A5256701, Gibco™, Waltham, MA, USA.) and 1% *v*/*v* penicillin-streptomycin (100 U/mL penicillin, 100 µg/mL streptomycin, Cat. No. 1TFS-1CC-15140122, Gibco™), and maintained at 37 °C in a humidified atmosphere containing 5% CO_2_. Cytotoxicity analysis was performed by seeding the cells at a density of 10,000 cells/well into 96-well plates (Cat. No. 165306, Thermo Fisher Scientific Inc., Waltham, MA, USA) and incubating for 24 h. Samples of 100 µL were added at two-fold serial dilutions (5 µg/mL CS^sol^, 5 mg/mL PE, 100 µg/mL LPS (Cat. No. 2630), and 10 µM CBD). Stock solutions of CS^sol^ (200 µg/mL CBD equivalent) and PE (500 mg/mL) were prepared. These were combined in ratios of 1:1 and 1:0.5, as shown in Appendix A. Final concentrations were selected based on cytotoxicity results of the individual components. Untreated cells served as the controls. While DMSO and PG (≤1% *v*/*v*) were used as vehicles, depending on the experimental condition. After incubation for 24 h, the media were removed, and 100 µL of 0.04 mg/mL MTT (Cat. No. 298-93-1) solution was added, and the plates were incubated for 2 h. in the dark. Formazan crystals were dissolved in DMSO, and the absorbance at OD_570_ was measured to calculate the cell viability:Cell viability (%) = (OD_570_ of sample)/(OD_570_ of control) × 100

#### 4.5.7. NO Inhibitory Assay and ELISA

Experimental analysis of the anti-inflammatory activity of the combination of CS and PE was adapted from previous studies [48,49]. Briefly, a total of 20,000 cells/well were seeded into 96-well plates and incubated for 24 h. Stock solutions of the combinations for the NO and ELISA assays were prepared in the same manner as used in the cytotoxicity assay. Subsequently, the cells were treated with 100 µL of a combination of PE:CS^sol^ or CBD, along with 1 µg/mL LPS, followed by incubation for 24 h. For NO measurement, 100 µL of the supernatant was mixed with 100 µL of Griess reagent (sulfanilamide (Cat. No. 63-74-1) and N-1-naphthylethylenediamine dihydrochloride (NED) (Cat. No. 1465-25-4), Tokyo Chemical Industry Co., Ltd., Tokyo, Japan) [50] and incubated for 10 min in the dark. The NO levels were measured at an OD_540_ absorbance and quantified by comparison to the NaNO_2_ (Cat. No. 7632-00-0, Tokyo Chemical Industry Co., Ltd.) standard curve.

Using ELISA kit (BioLegend^®^, San Diego, CA, USA), the supernatant collected from RAW 264.7 cells previously treated with LPS and the combination (as in the NO assay) was used to assess the levels of IL-8 (Cat. No. 431504), TNF-α (Cat. No. 430201), and IL-6 (Cat. No. 431304). Samples were diluted 1:10 with an appropriate diluent according to the manufacturer’s protocol (BioLegend^®^). Briefly, 96-well plates were coated with capture antibody and incubated overnight at 4 °C. After blocking, diluted samples and standards were added and incubated for 2 h at room temperature with shaking. The detection antibody and avidin-conjugated horseradish peroxidase were added sequentially, each followed by incubation and washing steps. TMB substrate (3,3′,5,5′-tetramethylbenzidine) was added, and the reaction was stopped with stop solution. Absorbance was measured at 450 nm using a microplate reader (CLARIOstar® Plus, BMG Labtech, Ortenberg, Germany).

#### 4.5.8. Statistical Analysis

Data are presented as mean ± standard deviation (SD) or standard error of the mean (SEM) from three independent experiments. Differences between the control and treatment groups were analyzed using one-way analysis of variance (ANOVA), with statistical significance set at *p* < 0.05.

## 5. Conclusions

PE is commonly used for sore throat relief and is recommended for bacterial infections, and CBD derived from *C. sativa* is known for its analgesic and anti-inflammatory properties. This study explores the combination of propolis and *C. sativa* extract, hypothesizing that an optimized combination may work additively to enhance the antibacterial and anti-inflammatory effects. The findings demonstrate that propolis and *C. sativa* extract exhibit antibacterial activity against pathogens such as *S. pyogenes* and *S. aureus*. Moreover, a combination of *C. sativa* solution and propolis solution at a ratio of 1:0.5 and 1:1 indicated that they exerted additive antibacterial effects. Notably, CBD and Δ9-THC were identified as active compounds against *S. pyogenes*, an important respiratory tract pathogen. In addition, the combination of *C. sativa* solution and propolis at a ratio of 1:0.5 at concentrations of 0.125 µg/mL and 625 µg/mL, respectively, significantly reduced NO generation and suppressed the release of proinflammatory cytokines (TNF-α, IL-6, and IL-8), thereby providing enhanced anti-inflammatory effects compared to the individual extracts. Overall, this novel combination shows significant advantages over existing treatments by enhancing both antibacterial and anti-inflammatory effects, suggesting its potential for further development in herbal pharmaceutical formulations. Given its local action and ease of administration, this combination may be considered promising for future formulation as an oral spray.

## Figures and Tables

**Figure 1 ijms-26-11181-f001:**
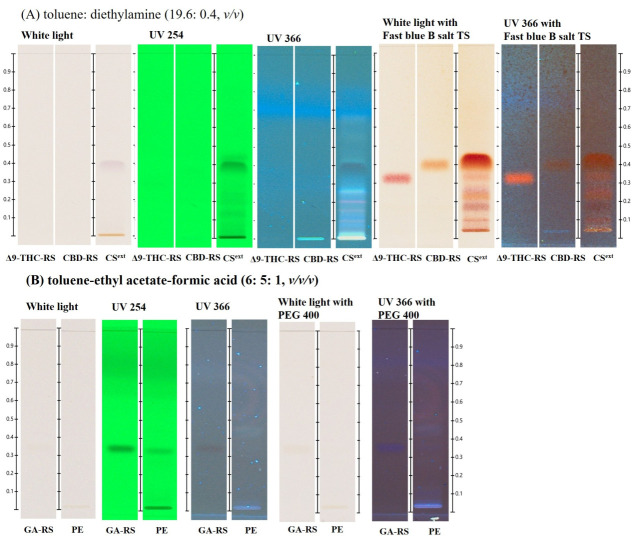
Chemical profile of *C. sativa* (CS^ext^) was compared with the reference standard materials 1 mg/mL cannabidiol (CBD) and 1 mg/mL Δ9-tetrahydrocannabinol (Δ9-THC) (**A**). The chemical profile of propolis extract (PE) was compared with the reference standard material 1 mg/mL gallic acid (GA) (**B**).

**Figure 2 ijms-26-11181-f002:**
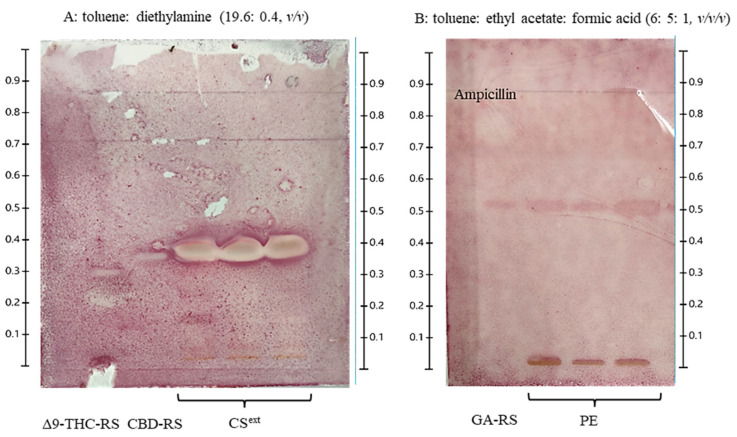
Thin-layer chromatography (TLC)-bioautography analysis of *Cannabis sativa* extract (CS^ext^) and propolis extract (PE) was performed by loading the samples onto a TLC plate under conditions similar to high-performance thin-layer chromatography (HPTLC), followed by incubation under disk diffusion conditions and staining with 0.2% (*w*/*v*) iodonitrotetrazolium chloride (INT) dye. Inhibition zones were observed at 10 mg/mL for CS^ext^ and at 1 mg/mL for the reference standards Δ9-THC and CBD (**A**), indicating antibacterial activity against *S. pyogenes*. In contrast, no inhibition zones were observed at 100 mg/mL for PE or at 1 mg/mL for gallic acid (GA) as the reference standard (**B**).

**Figure 3 ijms-26-11181-f003:**
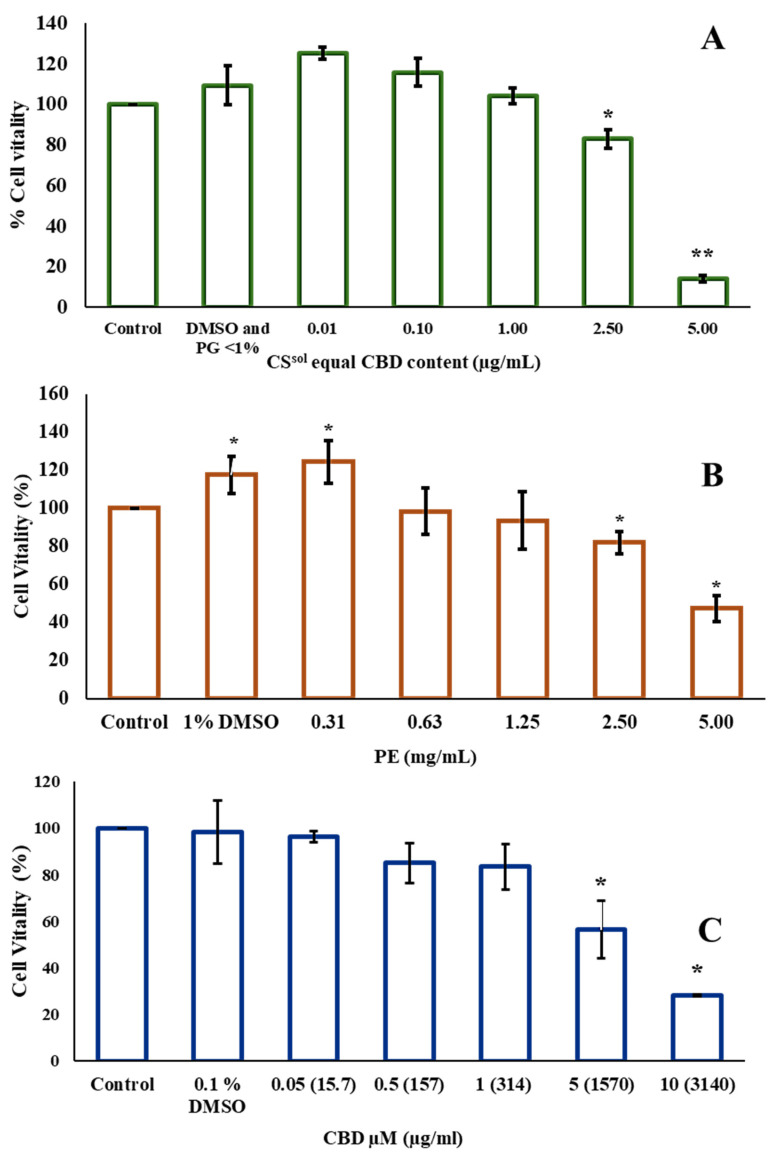
Viability of RAW264.7 cells after treatment with propolis extract (PE) (**A**), eq. cannabidiol (CBD) content in *Cannabis sativa* solution (CS^sol^) (µg/mL) (**B**), and CBD µM (µg/mL) (**C**) for 24 h. The viability of the treated cells was determined using an MTT assay (*n* = 3). Results are presented as the mean ± standard error of the mean (SEM) and analyzed using a one-way ANOVA. ** p* < 0.05 and *** p* < 0.01 when compared with the untreated group.

**Figure 4 ijms-26-11181-f004:**
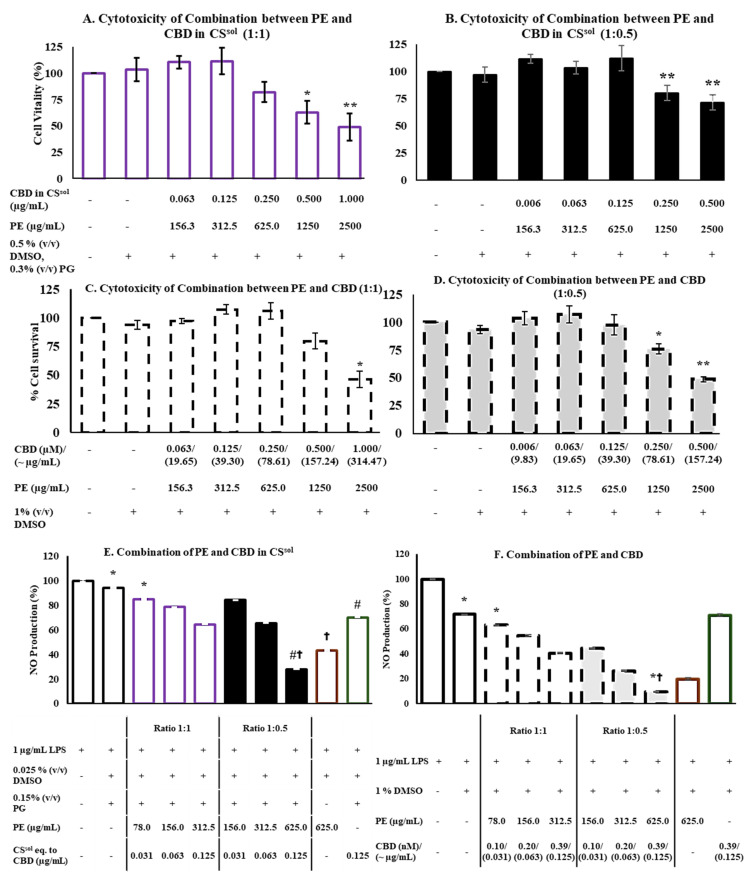
The viability of RAW264.7 cells in Combination 1 of propolis extract (PE) and cannabidiol (CBD) content in *C. sativa* solution (CS^sol^) at a ratio of 1:1 (**A**) and 1:0.5 (**B**) and in Combination 2 of PE and CBD at a ratio of 1:1 (**C**) and 1:0.5 (**D**) was determined using the MTT assay (*n* = 3). Results are presented as a percentage of viability (means ± standard error of the mean (SEM)), and a one-way ANOVA was used to determine significant differences. * *p* < 0.05 and ** *p* < 0.01 compared to the untreated group. NO production was measured from the supernatant of RAW264.7 cells after stimulation with LPS and after treatment with samples: Combination 1 of PE and CS^sol^ eq. to CBD (**E**), and Combination 2 of PE and CBD (**F**) using Griess reagent (*n* = 3). The results are shown as the level of NO production (means ± SEM), and a one-way ANOVA was used to determine significant differences. * *p* < 0.05 when compared with the LPS-treated group; # *p* < 0.05 when comparing with the LPS + 625 µg/mL PE-treated group; and † *p* < 0.05 when comparing with the LPS + CS^sol^ eq. to CBD 0.125 µg/mL or LPS + CBD 0.125 µM-treated groups.

**Figure 5 ijms-26-11181-f005:**
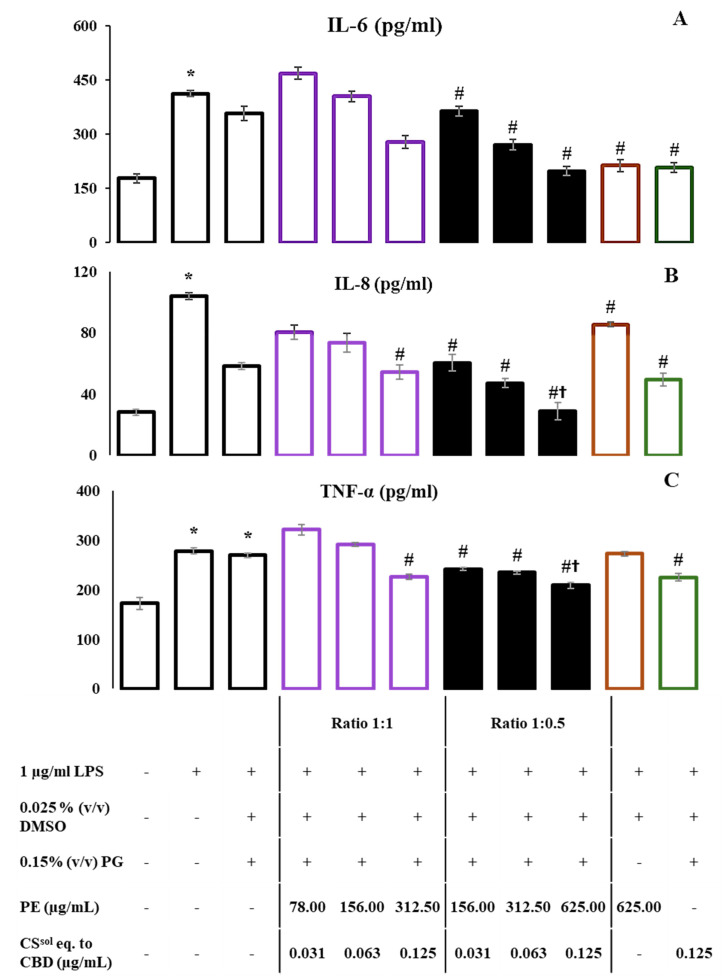
Cytokine levels (IL-6; (**A**), IL-8; (**B**), and TNF-α; (**C**)) after stimulation with 1 µg/mL lipopolysaccharide (LPS) and the combinations of propolis extract (PE): *C. sativa* solution (CS^sol^) eq. to cannabidiol (CBD) in ratios of 1:1 and 1:0.5 are represented as the mean ± standard error of the mean (SEM), and a one-way ANOVA was used to determine significant differences. * *p* < 0.05 when compared with the untreated group; # *p* < 0.05 when compared with the LPS-treated group; and † *p* < 0.05 when compared with the LPS + 6.25 µg/mL of PE-treated group.

**Table 1 ijms-26-11181-t001:** Antibacterial activity of propolis extract solution (PE^sol^) and *C. sativa* extract (CS^ext^) using the agar disk diffusion method.

Samples (Concentration/disc)	Zone of Inhibition (mm) (Means ± Standard Deviation)			
	*S. pyogenes* DMST 4369	*S. aureus* ATCC 25923	*P. aeruginosa* ATCC 9027	*K. pneumoniae* ATCC 13883
100% *v*/*v* PE^sol^	12.2 ± 0.4	13.5 ± 1.1	14.7 ± 0.6	0.0 ± 0.0
20% *v*/*v* PE^sol^	12.0 ± 0.0	13.0 ± 0.0	15.0 ± 1.7	0.0 ± 0.0
0.4 mg CS^ext^ (eq. to 229 µg CBD)	17.0 ± 1.0	13.0 ± 0.0	0.0 ± 0.0	0.0 ± 0.0
10% *v*/*v* DMSO; negative control	0.0 ± 0.0	0.0 ± 0.0	0.0 ± 0.0	0.0 ± 0.0
Sterile water negative control	0.0 ± 0.0	0.0 ± 0.0	0.0 ± 0.0	0.0 ± 0.0
30 µg chloramphenicol; positive control	28.0 ± 0.0	n.d.	16.0 ± 0.0	30.0 ± 0.0
200 µg amoxicillin; positive control	28.0 ± 0.0	29.4 ± 0.0	12.5 ± 0.0	17.5 ± 0.0

n.d.: not determined; CBD, cannabidiol; DMSO, dimethyl sulfoxide.

**Table 2 ijms-26-11181-t002:** Minimum inhibitory concentration (MIC) and minimum bactericidal concentration (MBC) of propolis extract solution (PE^sol^) and *C. sativa* solution (CS^sol^).

Microorganisms	PE^sol^ (% *v*/*v*)		CS^sol^ Eq. to CBD (µg/mL)	
	MIC	MBC	MIC	MBC
Gram-positive bacteria				
*S. aureus* ATCC 25923	2.5	5.0	1.6	3.1
*S. pyogenes* DMST 4369	2.5	5.0	1.0	2.0
Gram-negative bacteria				
*P. aeruginosa* ATCC 9027	5.0	10.0	40.0	80.0
*K. pneumoniae* ATCC 13883	1.3	2.5	250.0	500.0

CBD, cannabidiol.

**Table 3 ijms-26-11181-t003:** Result of the checkerboard assay with the fractional inhibitory concentration (FIC) and FIC index of the two combinations between propolis extract solution (PE^sol^) (% *v*/*v*) and *Cannabis sativa* solution (CS^sol^) eq. to cannabidiol (CBD) (µg/mL).

Bacteria Strain	MIC	PE^sol^ (% *v*/*v*)	CS^sol^ Eq. to CBD (µg/mL)	FIC Index	Interpretation
Gram-positive bacteria					
*S. aureus* ATCC 25923	MIC of single compounds	2.50	1.56	1.50	Indifference
	MIC of Combination 1	1.25	1.56		
	FIC	0.50	1.00		
	MIC of single compounds	2.50	1.56	1.00	Additive
	MIC of Combination 2	1.25	0.78		
	FIC	0.50	0.50		
*S. pyogenes* DMST 4369	MIC of single compounds	2.50	1.00	1.00	Additive
	MIC of Combination 1	1.25	0.50		
	FIC	0.50	0.50		
	MIC of single compounds	2.50	1.00	1.50	Indifference
	MIC of Combination 2	2.50	0.50		
	FIC	1.00	0.50		
Gram-negative bacteria					
*P. aeruginosa* ATCC 9027	MIC of single compounds	5.00	40.00	1.50	Indifference
	MIC of Combination 1	5.00	20.00		
	FIC	1.00	0.50		
	MIC of single compounds	5.00	40.00	1.25	Indifference
	MIC of Combination 2	5.00	10.00		
	FIC	1.00	0.25		
*K. pneumoniae* ATCC 13883	MIC of single compounds	1.25	250.00	4.00	Indifference
	MIC of Combination 1	2.50	500.00		
	FIC	2.00	2.00		
	MIC of single compounds	1.25	250.00	3.00	Indifference
	MIC of Combination 2	2.50	250.00		
	FIC	2.00	1.00		

MIC, minimum inhibitory concentration.

## Data Availability

The original contributions presented in this study are included in the article and Appendix A. Further inquiries can be directed to the corresponding author.

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
