# Peer review of "Enhanced Antibacterial and Anti-Inflammatory Activities of the Combination of Cannabis sativa and Propolis Extracts: An In Vitro Study"

_ijms, 2025, doi:10.3390/ijms262211181_

Round 1
Reviewer 1 Report
Comments and Suggestions for Authors
Authors in the manuscript ”Promising Combination of Cannabis sativa and Propolis Extracts with Antibacterial and Anti-inflammatory Properties for Sore Throat Treatment” report the antibacterial and antiinflammatory effects fo combination of these extracts. However, connecting these in vitrro results directly to sore throat treatment seems to be slightlyu misleading.
Both CS extract and propolis are proved alredy for their antibacterial and antiinflammatory effects in various in vitro and in vivo studies. What are the new areas that authors are trying to explore or what new information/evidence is obtained, should be clearly mentioned.
Authors should compare the effects of combination with individual agents (CS extract and propolis) and see if there is some synergestic effects. Evaluating combination only does not provide idea about their possible synergestic and/or antagonistic activities.
Reviewer 2 Report
Comments and Suggestions for Authors
Perstwong et al.' work is devoted to studying the antibacterial and anti-inflammatory potential of Cannabis sativa and propolis extracts as alternative treatments for bacterial sore throat, hypothesizing a synergistic effect and aiming to optimize their combined dosage.
Comments
- In introduction, clearly state the novelty and importance of this work with respect to the existing knowledge in literature
- Figure 3,4 and 5 should be enhanced the resolution and size in the present form in not clear for reading.
- Cytotoxicity was assessed only on RAW 264.7 murine cells; it would be advisable to evaluate toxicity on healthy human cell lines such as MRC-5 (human fibroblasts). Additionally, toxicity should be examined using another model, such as zebrafish. The duration of macrophage cell treatment is not mentioned anywhere.
- The authors monitored the cytotoxicity of pure CBD and a Cannabis extract containing CBD at a defined concentration (Figure 3). Based on the results, it appears that both pure CBD at 314 µg/mL and the Cannabis extract containing 2.5 µg/mL of CBD exert a similar effect on macrophage viability, approximately 80%. Clarification is requested regarding this observation.
- The discussion section is overly descriptive. Several sentences would be more appropriately placed in the Results section to improve clarity and structure.
Reviewer 3 Report
Comments and Suggestions for Authors
This manuscript by Vimolmangkang and coworkers investigates the antibacterial and anti-inflammatory effects of combining Cannabis sativa extract (CS) and propolis extract (PE) as a potential treatment for bacterial-induced sore throat. The study's hypothesis that the combination would exhibit enhanced effects was largely supported, showing an additive antibacterial effect against Gram-positive bacteria. Furthermore, the combination significantly reduced nitric oxide (NO) production and suppressed pro-inflammatory cytokines (TNF-, IL-6, and IL-8) in LPS-stimulated macrophages, demonstrating superior anti-inflammatory activity compared to individual extracts. The methodology is good, employing checkerboard assays, HPTLC, and ELISA. The work is novel and relevant to natural product research.
Here are my key comments and suggestions for revision:
1) The concentrations chosen for the NO and cytokine assays in Combination 1 (PE and CS-sol equivalent to CBD) appear to be the highest non-toxic concentrations tested in the cytotoxicity assay (PE 625 μg/mL and CS-sol eq. to CBD 0.125 μg/mL in Figure 4A-B). Yet, the manuscript states that those concentrations (PE 2.5 mg/mL and CS-sol eq. to CBD 1 μg/mL in Figure 3A-B) were selected for the combinations. The authors must clarify and rationalize the final concentrations used in Figure 4E and Figure 5. It is critical to confirm if the concentrations 625 μg/mL PE and 0.125 μg/mL CS (CBD eq.) were chosen because they represent a significant anti-inflammatory effect (effective dose) or simply the maximum non-toxic dose (limit of toxicity). Clarification is needed, as 625 μg/mL PE is one-quarter of the non-toxic limit in Figure 3A (2.50 mg/mL).
2) In Figure 3A, The 5.00 mg/mL PE concentration shows a significant decrease in cell viability (**p<0.01), but the control viability is only ~100%. However, in Figure S2A, the solvent propylene glycol (PG) shows a significant reduction in viability (**p<0.01) at a lower concentration of 0.5% v/v. The relationship between PE-sol and its vehicle (DMSO and PG) needs clarification, as PE-sol was resuspended in DMSO for the anti-inflammatory assay. The final volume percentages of all solvents (DMSO and PG) used in the cytotoxicity assay must be stated clearly, especially for PE 5.00 mg/mL and the subsequent combination assays from Figure 4.
3) The TLC-bioautography assay showed no antibacterial activity for PE or the Gallic Acid (GA) reference standard against S. pyogenes. This contradicts the main results that exhibits antibacterial activity against S. pyogenes with a zone of 12.0 mm inhibition of and of 2.5% v/v.
4) In the abstract, clarify the specific combination ratio that showed the best anti-inflammatory effect. The conclusion states the anti-inflammatory effect was "superior" for the combination, but the most significant suppression was seen at one specific ratio: PE:CS-sol (CBD eq.) at 625:0.125 μg/mL.
5) The Preparation Steps for Combination 1 (Ratio 1:1) in Figure S1 shows concentrations for the MIC-MBC assay and the Cytotoxicity assay. Authors should clarify that the Cytotoxicity assay preparation is the basis for the Anti-inflammatory assay as well, as suggested by the text from manuscript.
6) Table 2 lists the units for PE-sol as %v/v and CS-sol as μg/mL (eq. to CBD). However, in the Discussion, PE's MIC is also reported in mg/mL (MIC of 2.5%v/v is equivalent to 25 mg/mL).
7) In the Discussion (lines 363-380), the authors reference a study on the CBD/moringin combination that improved inflammatory and oxidative markers and anti-apoptotic properties. Authors should briefly mention whether the PE/CS combination in this study is hypothesized to share a similar, possibly multi-target mechanism, in line with the dual nature of the extracts, rather than just listing external findings.
8) The PE:CS-sol combination at PE 625 μg/mL and CS-sol 0.125 μg/mL shows a significant reduction in all three cytokines (IL-6, IL-8, and TNF-) compared to the -treated group (#p<0.05). However, for IL-8, this same concentration is also significantly lower than the 625 μg/mL PE alone group (†p < 0.05).
9) The method states the CS-ext solution stock was prepared to contain 5mg/mL CBD. However, the agar disk diffusion method used a CS-ext concentration of 20 mg/mL with 11.44mg/mL CBD. The CS-ext purity is 57.20w/w CBD. The stock concentration in Section 4.2 should be re-examined. It seems the CS-sol stock concentration given for the MIC/MBC assay (5 mg/mL CBD) is inconsistent with the CS-ext concentration given in the Agar Disk Diffusion section (11.44 mg/mL CBD).
Reviewer 4 Report
Comments and Suggestions for Authors
The manuscript titled “Promising combination of Cannabis sativa and propolis extracts with antibacterial and anti-inflammatory properties for the treatment of sore throat” presents a study exploring the potential synergistic effects of Cannabis sativa and propolis extracts as an alternative therapeutic approach for sore throat. The topic is current and relevant, given the increasing need for natural products with antimicrobial and anti-inflammatory activity in the context of antibiotic resistance and adverse drug effects.
The study aims to provide scientific evidence to support a new formulation of medicinal cannabis with improved antibacterial and anti-inflammatory properties. Overall, the research has potential scientific and practical value; however, several methodological and reporting aspects require clarification and improvement to strengthen the validity and reproducibility of the results.
- Please define all abbreviations (e.g. CSext , PE , GA , etc. ) at their first occurrence in the text to improve clarity for readers unfamiliar with these terms.
It should be clearly stated who performed the HPLC analysis of THC and CBD content - the supplier ( Leapdelab Co., Ltd. ) or the authors. This clarification is important for transparency and reproducibility.
The manuscript would benefit from including quantitative analyses of the extracts (e.g. total phenolic content, flavonoid content, or concentration of key compounds) to better characterize the materials used.
The manuscript mentions “additive antibacterial effects,” but the method used to determine the type of interaction (additive, synergistic, or antagonistic) is not described.
Authors are encouraged to indicate how this combination could be practically applied—for example, as an oral spray, lozenge, mouthwash, or other suitable pharmaceutical form—to clarify the potential route of administration and therapeutic use.
Round 2
Reviewer 2 Report
Comments and Suggestions for Authors
The authors have incorporated the reviewer’s suggestions and significantly improved the manuscript, which is now suitable for publication in the journal. A minor remaining issue concerns the uniformity of the 'ug' (µg) notation throughout the text.
Author Response
Comments 1: The authors have incorporated the reviewer’s suggestions and significantly improved the manuscript, which is now suitable for publication in the journal. A minor remaining issue concerns the uniformity of the 'ug' (µg) notation throughout the text.
Response 1: Thank you very much for your valuable time and feedback on this manuscript. We have carefully reviewed the entire text and revised all instances of the unit from ug to µg to ensure consistency. These corrections are clearly marked in red in the revised manuscript.
Reviewer 3 Report
Comments and Suggestions for Authors
All previously raised concerns have been addressed in the revisions; the manuscript is now suitable for publication.
Author Response
Summary: We sincerely appreciate your time and effort in reviewing our manuscript. Please find below our detailed responses to your comments, along with corresponding revisions highlighted in red text in the re-submitted files.
Thank you very much for your positive feedback. We have thoroughly improved the introduction section and figures as suggested.
Reviewer 4 Report
Comments and Suggestions for Authors
Accept in present form
Author Response
Comments 1: Accept in present form
Response 1: Thank you very much for your positive feedback.